# Microbial dysbiosis reflects disease resistance in diverse coral species

Nicholas J. MacKnight[1], Kathryn Cobleigh[2], Danielle Lasseigne[2], Andia Chaves-Fonnegra[2,3], Alexandra Gutting[2,4], Bradford Dimos[1], Jendahye Antoine[2], Lauren Fuess[1,5], Contessa Ricci[1], Caleb Butler[1], Erinn M. Muller[6], Laura D. Mydlarz[1] & Marilyn Brandt [2✉]

Disease outbreaks have caused significant declines of keystone coral species. While forecasting disease outbreaks based on environmental factors has progressed, we still lack a comparative understanding of susceptibility among coral species that would help predict disease impacts on coral communities. The present study compared the phenotypic and microbial responses of seven Caribbean coral species with diverse life-history strategies after exposure to white plague disease. Disease incidence and lesion progression rates were evaluated over a seven-day exposure. Coral microbiomes were sampled after lesion appearance or at the end of the experiment if no disease signs appeared. A spectrum of disease susceptibility was observed among the coral species that corresponded to microbial dysbiosis. This dysbiosis promotes greater disease susceptiblity in coral perhaps through different tolerant thresholds for change in the microbiome. The different disease susceptibility can affect coral's ecological function and ultimately shape reef ecosystems.

[1] University of Texas at Arlington, Arlington, TX, USA. [2] University of the Virgin Islands, St. Thomas, VI, USA. [3] Florida Atlantic University, Harbor Branch Oceanographic Institute/Harriet L.Wilkes Honors College, Fort Pierce, FL, USA. [4] The Nature Conservancy, Dallas, TX, USA. [5] Texas State University, San Marcos, TX, USA. [6] Mote Marine Laboratory, Sarasota, FL, USA. ✉email: mbrandt@uvi.edu

Disease is a natural force in ecosystems and at low prevalence will shape species evolution over time[1,2]. In recent decades, stressors on ecosystems driven by climate change, habitat loss and alteration, and globalization have increased disease prevalence, in some cases leading to devastating outbreak events in wild populations[3–5]. These outbreaks have reshaped entire ecosystems, both terrestrial and marine[6]. Marine disease outbreaks have driven foundational species to endangerment, including California abalone, West coast sea star, and elkhorn and staghorn corals in the Caribbean[7–10]. Disease affects populations and communities globally and quantifying the impact among species will provide predictive insight into the changing functional ecology of these ecosystems.

White plague disease (WPD) is one of the most destructive diseases in the Caribbean[11,12] affecting a large number of coral species and reducing the biodiversity and function of reef ecosystems[13–16]. WPD has been described as affecting Caribbean corals since the 1970s and is characterized by lesions originating at the base of the colony and expanding rapidly, resulting in significant partial and total mortality to affected colonies[17]. WPD is a suspected bacterial infection[17], though there has been considerable debate as to whether WPD represents one or more etiologies[18]. Recent laboratory experimentation has confirmed species-specific susceptibility in response to exposure to WPD[19], however, it is unknown what is driving these species differences in susceptibility. Traits that could influence disease susceptibility include immune capacity, life-history strategies, and coral-associated microbial communities[20].

As with many coral diseases, the causative agent of WPD remains unknown. Evidence among studies points to individual bacteria[21], possible polymicrobial origins[22], and even viral pathogens[23], all of which question the traditional view of a singular pathogenic etiology[24,25]. In human disease studies, there is growing literature on microbiome imbalance or dysbiosis that is responsible for disease etiology[26]. In coral diseases, microbiome shifts or dysbiosis also may be more appropriate than the one-pathogen-one disease concept[24,25,27–33]. By measuring microbial dysbiosis, we can evaluate thresholds of microbial change before host and microbiome symbiosis break down into a diseased state, consider coral disease etiology beyond the singular pathogen hypothesis, monitor the compounding effects of multiple stress events, and predict coral species survival likelihood[34–37].

To investigate the relationship between disease susceptibility and microbial community responses to WPD exposure, this study simultaneously characterized the phenotypic and microbial responses of seven Caribbean coral species when exposed to WPD in a controlled laboratory experiment. The seven species represented diverse life-history strategies and roles within Caribbean reef ecosystems[38]. We identified differences among coral species not only in their phenotypic responses to WPD exposure but in their microbial responses as well. We then referred to the literature for any known functional roles and relevance in coral studies of individual bacteria for their potential role as disease-associated or disease-preventing bacteria based on their abundances among treatments and the treatment outcomes. Understanding the link between microbial shifts and disease susceptibility can help identify the potential mechanisms driving disease resistance, which will assist in predicting future coral assemblages.

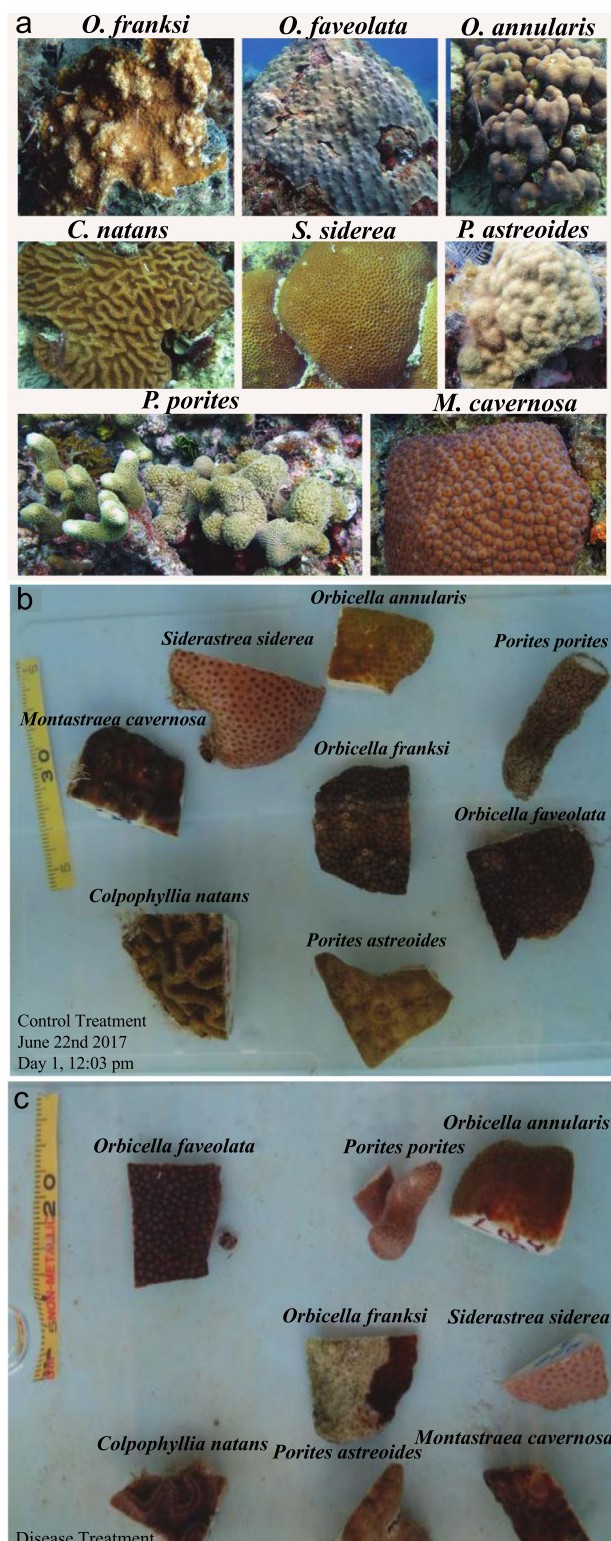

## Results

**Disease prevalence.** Disease prevalence significantly differed among species (Fig. 1a) (Fisher's Exact Test: $p = 0.0074$) and ranged from 0 to 100% (Fig. 2a). Six of the seven species tested had fragments that exhibited progressive lesions indicative of WPD. *O. faveolata* was identified as the most susceptible species with 100% affected. Only *M. cavernosa* did not show signs of lesion development in any fragment exposed to WPD. No lesions formed under control conditions in any species. Pairwise comparisons identified significantly different disease prevalence between species pairs (Supplementary Table 1).

**Disease severity.** Disease progression rates were significantly different among species (df = 5, $X^2 = 28.627$, $p < 0.0001$). *O.*

**Fig. 1 Coral species and experimental design. a** *Orbicella faveolata, Orbicella annularis, Orbicella franksi, Colpophyllia natans, Montastraea cavernosa, Porites astreoides, Porites porites, Siderastrea siderea.* All photos by M. Brandt except *M. cavernosa* by T.B. Smith. Disease transmission experimental design for (**b**) control and (**c**) disease treatment. Note the healthy *O. franksi* in the center of the control and the white plague-infected *O. franksi* in the center of the disease treatment. There were five aquaria assigned as control treatments and five aquaria assigned as disease treatments. Every tank included one fragment from each of the seven coral species. When a coral colony was collected, it was split in two, with one fragment designated for control treatment and the other paired fragment going to a disease exposure treatment.

---

*annularis*, which had the fastest lesion progression rate had a significantly different lesion progression rate than *C. natans* and *S. siderea* (Supplementary Table 4). Corals in the genus *Orbicella* had the highest relative lesion progression rate (Fig. 2b), with *O. annularis* and *O. faveolata* exhibiting the first and second fastest average progression rates (15.75 cm$^2$ day$^{-1}$ and 13.79 cm$^2$ day$^{-1}$, respectively). *C. natans* and *S. siderea* showed slower progression rates (3.85 cm$^2$ day$^{-1}$ and 2.60 cm$^2$ day$^{-1}$, respectively), while the only *P. astreoides* and *P. porites* colonies to exhibit disease lesions had progression rates that were considered intermediate (9.37 cm$^2$ day$^{-1}$) or slow (2.03 cm$^2$ day$^{-1}$), respectively.

**Disease incidence**. Throughout the seven-day disease exposure, *O. faveolata* fragments and *O. annularis* fragments presented lesions characteristic of WPD between days 5 and 6 of exposure (Fig. 3a). *C. natans* showed disease signs between 2 to 6 days of exposure, and *S. siderea* showed disease signs between days 4 and 5 of exposure (Fig. 3b). Both *P. porites* and *P. astreoides* had disease incidence on day 6 of exposure. None of the *M. cavernosa* fragments showed signs of WPD over the course of the experiment (Fig. 3c).

**Relative risk**. The relative risk of developing WPD signs after being exposed to a WPD coral differed among species (Fig. 4). *O. faveolata, O. annularis,* and *C. natans* all had significant risks of infection (Fig. 4). The median risk of these three species was around a value of 9, indicating the likelihood of developing WPD was 9 times greater after exposure. *S. siderea, P. porites,* and *P. astreoides* did not exhibit significantly increased risk of disease after exposure to WPD, although their median risk values were still elevated above 1 (ranging from 3 to 8) suggesting a higher overall risk to WPD after exposure. *M. cavernosa* had an estimated relative risk close to 1.0, indicating no elevated disease risk from exposure to WPD.

**Microbiome**

*Bacterial community dissimilarity and dysbiosis.* Microbiomes significantly differed among the seven[7] coral species (PERMANOVA: $F = 3.91$, $p < 0.001$) (Fig. 5) and were also significantly different between control and disease treatments ($F = 3.3381$, $p = 0.0045$). When fragments were grouped by treatment outcomes (i.e., control, disease-exposed, disease-infected), significant differences were also detected in the microbial communities ($F = 1.97$, $p = 0.0195$), but pairwise comparisons showed a significant difference at $p < 0.1$, but not at the Bonferonni corrected $p$-value of $p = 0.017$ (Supplementary Table 5). These results indicate shifts in the relative abundances of the microbes once an individual was exposed to WPD or became diseased (Table 2). No interaction was detected between species and treatment ($F = 0.94$, $p = 0.5714$), or between species and treatment outcome ($F = 1.1026$, $p = 0.235576$) suggesting consistent differences among coral species. To look further into how each coral species'

microbial community was changing based on treatment outcome, the overall dissimilarity was compared within species relative to the healthy control state (Table 1). Bacterial community dissimilarity was consistently higher in disease-infected fragments than in disease-exposed fragments relative to their paired controls (Table 1). Coral species that remained disease-exposed and did not contract the disease were more dissimilar to each other relative to the control condition dissimilarity (Table 2). While inversely, overall bacterial community similarity was greater in coral that became disease-infected. Notably, *M. cavernosa*, the only species to not exhibit disease signs, did not show overlap with any other species in the NMDS ordination (Fig. 5). All other species showed some overlap with the others regardless of treatment outcome.

*Treatment outcome-specific bacterial communities.* Of the 7225 unique OTUs, 1243 OTUs had greater than 97% sequence similarity to reference sequences allowing for species-level classification of 1243 unique bacterial species identified in this study, 29 bacteria represented 70% of the microbiome across all samples and were identified by being in 3% abundance or higher in either control, disease-exposed, or disease-infected states. Five of these 29 bacteria significantly differed in their relative abundance between treatment outcomes. Comparisons between the control, diseased-exposed, and disease-infected among all 7 coral species combined showed large overall differences of these five bacteria (Fig. 6a). *Nautella italica, Pseudoalteromonas sp.*, and *Thalassobius mediterraneus* displayed low relative abundance in the control treatment but were significantly higher in disease-infected fragments (Fig. 6c). Conversely, *Endozoicomonas spp.* and *Burkholderia ubonesis* showed the highest abundance in disease-exposed treatments and low or no abundance in the disease-infected (Fig. 6b).

*Species-specific commensal or beneficial bacteria.* Certain microbes were consistently present among fragments within a coral species but absent or in low abundance in other species suggesting an identifiable species–specific microbiome (Fig. 7). The bacterial family Hahellaceae, consisting of *Endozoicomonas sp.*, was notably dominant in *P. astreoides*, but not detectable in any great abundance in the other coral species tested. Burkholderiaceae and Spirochaetaceae were also highly abundant in *M. cavernosa* and *O. faveolata*, respectively, and not in other species. Micro-coccacaea dominated the microbiomes of *O. annularis* and *M. cavernosa* (*Anthrobacter* species; Supplementary Fig. 1). Within this family specifically, *Anthrobacter ramosus* composed a large proportion of the Micrococcacaea identified (20–92% in *M. cavernosa* and 49–80% in *O. annularis*).

*Microbial diversity and richness.* Alpha and beta diversity of the bacterial community significantly differed among species and species × treatment outcome, but not between treatments, or among treatment outcome (Table 3, Fig. 8). While alpha diversity (Fig. 8a) showed no trends related to species susceptibility, beta diversity (Fig. 8b) in disease-exposed fragments was significantly different among species and was reduced in the highly affected coral species (*O. annularis, C. natans*) and intermediately affected species (*S. siderea*), while no change in beta diversity was observed between control and disease-exposed fragments in low susceptible species (Fig. 8c, d, Table 3).

**Discussion**

This study delineated surface microbial responses of seven Caribbean reef-building coral species exposed to WPD. We found a gradient of disease susceptibility that was reflective of microbial community responses. Based on their phenotypic responses to

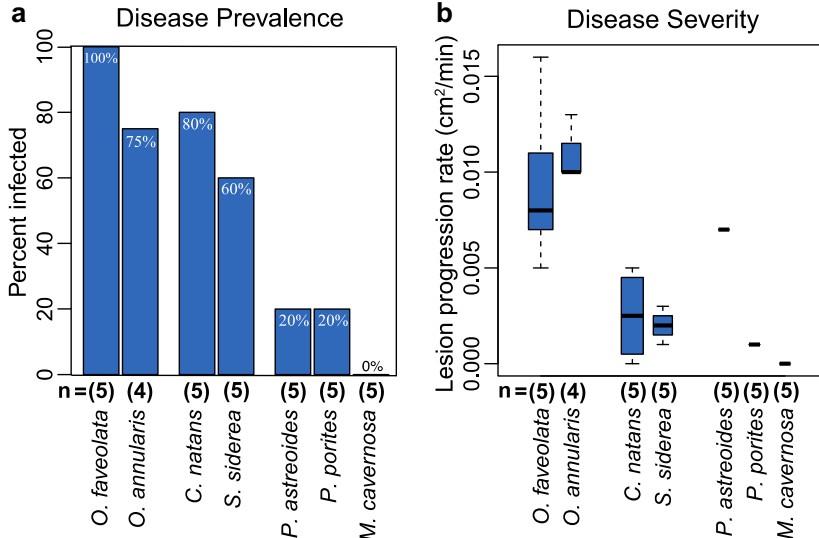

**Fig. 2 Disease prevalence and severity. a** Disease prevalence (percentage of replicates that contracted the disease) for each species. **b** Disease severity (the rate at which the disease lesion progressed across the infected coral after contracting white plague disease); only the corals that contracted the disease had their lesion progression rates graphed.

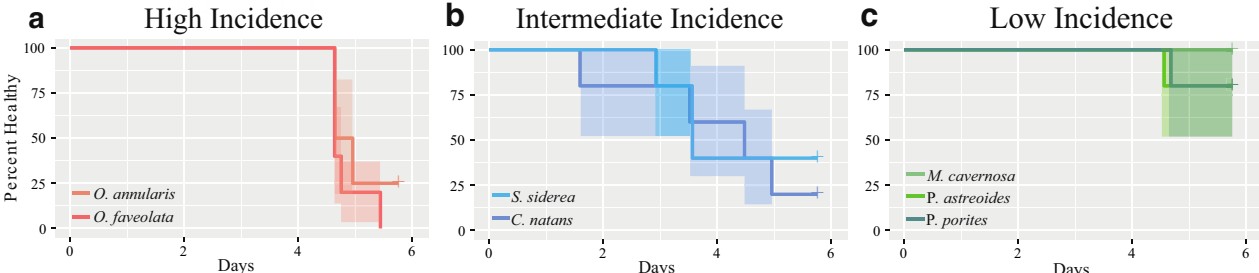

**Fig. 3 Disease incidence.** Survival curves showing the proportion of corals that remained healthy (i.e., did not develop lesions) over the course of the experiment for species that showed high (**a**), intermediate (**b**), and low (**c**) incidence of disease. Shading around survival curves represents 95% confidence intervals.

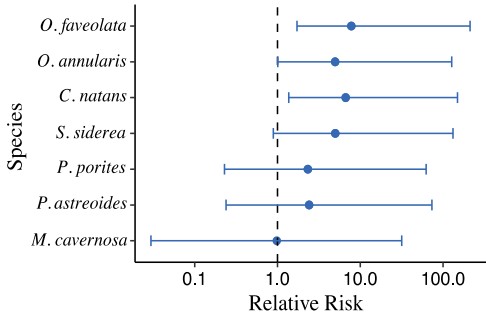

**Fig. 4 Relative risk.** The median relative risk the species will contract white plague disease with 95% credible intervals.

disease exposure, the seven species fell into three groups: (1) highly susceptible (*O. faveolata* and *O. annularis*) characterized by high disease prevalence and fast lesion progression rates, (2) intermediate susceptible (*C. natans* and *S. siderea*) characterized by high disease prevalence and slow lesion progression rates, and (3) low susceptible or tolerant (*P. astreoides*, *P. porites*, and *M. cavernosa*) characterized by low to no disease prevalence and slow lesion progression rates.

As seen in the field[11,39] and in experimental studies[19], *Orbicella* species displayed the highest disease susceptibility. As major structural reef builders, the high susceptibility of orbicellids has the potential to shift the physical growth and function of coral reefs, which has already been seen in many Caribbean locales[12,40,41]. Not only were individuals of both orbicellid species highly susceptible to disease in this study, but they also showed the highest severity of the disease. *P. astreoides*, *P. porites*, and *M. cavernosa* were relatively resistant to WPD. Each of these species has historically been documented as stress-tolerant or weedy[38] and their relative abundances are currently increasing as overall coral cover declines[42]. As coral disease outbreaks become more common and severe because of continued degrading local conditions and the exacerbating effects of climate change, disease-resistant species will likely dominate Caribbean reefs.

To capture the expected disease incidence when WPD was present, the relative risk of infection was determined for each species in this study. From this, *O. faveolata*, *O. annularis*, and *C. natans* each showed an elevated significant risk of contracting WPD if exposed. This pattern may have been related to the phylogenetically similar *O. franksi* serving as the source coral for the experiments, thereby more easily transmitting to the other orbicellids. However, *M. cavernosa* was the least susceptible to WPD exposure yet is closest in phylogeny to the orbicellids[43]. In

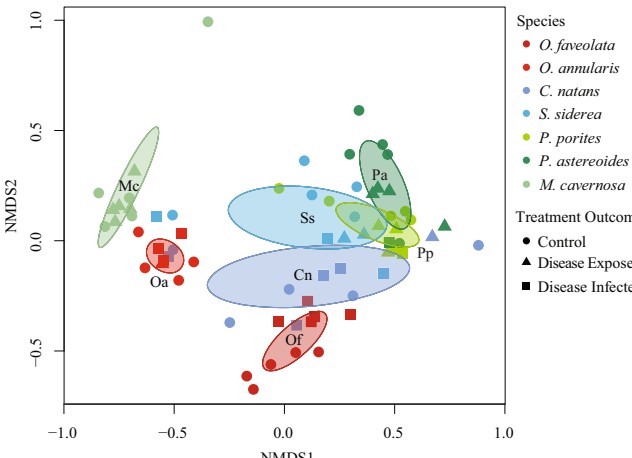

**Fig. 5 NMDS of Bacterial Communities.** Non-metric multidimensional scaling (NMDS) (Stress = 0.13) of bacterial communities from the tissue of the seven species tested organized by species and treatment outcome (control, disease-exposed, disease-infected). Ovals represent 95% confidence intervals.

**Table 1 Dysbiosis.**

| Species | Disease-exposed vs. control | Disease-infected vs. control |
|---|---|---|
| O. faveolata | – | 0.59 |
| O. annularis | 0.188 | 0.302 |
| C. natans | 0.73 | 0.762 |
| S. siderea | 0.701 | 0.746 |
| P. porites | 0.762 | 0.912 |
| P. astreoides | 0.393 | 0.716 |
| M. cavernosa | 0.381 | – |

The values are the overall dissimilarity of microbial communities calculated by SIMPER analysis for fragments between treatment outcomes within each species. Values closer to 1 indicate higher microbial dissimilarity, while values closer to 0 represent similar microbial communities.

addition, the susceptibility patterns we observed in our transmission experiment corresponded with field data that show these species (*O. faveolata*, *O. annularis*, and *C. natans*) clustered together because of high disease levels following 2005 and 2010 bleaching that resulted in significant mortality[44]. This independent relative risk calculation reiterates what we have observed from other coral diseases and that our laboratory-based experiments are consistent with field observations, making them an essential tool for predicting species assemblages in the future.

The origins of many coral diseases, including WPD, is still unknown and there are several hypotheses regarding how and why corals become diseased. For instance, the coral disease can originate from a foreign individual pathogen introduction and spread from coral to coral[45,46] or from microbiome changes that cause dysbiosis in the commensal surface microbial community[47]. Disease phenotypes may also be the result of secondary infection following an extreme stress event or increased bacterial virulence or pathogenicity following some disturbance[12,48,49]. Our experiment and dataset provide a unique opportunity to explore some of these hypotheses in seven species of coral to determine consistencies and identify what is unique in susceptible and resistant species.

The bacterial community in diseased fragments were consistently more similar among species relative to any other treatment outcome (as seen in Table 2). This response suggests that diseased fragments had similar microbial constituents while in the diseased state, which could reflect a community of pathogenic or opportunistic bacteria. This microbial convergence is likely the result of disease exposure rather than tank effects because diseased fragments started showing signs of tissue loss at different time points in the study and were removed from tanks based on differing fragment lesion development. In addition, the microbiomes of fragments that were exposed to disease, but stayed apparently healthy became more dissimilar than in the controlled state. To further explore the convergent microbiome in disease-infected fragments, bacteria increasing in relative abundance in this disease-infected state were identified.

*N. italica*, *T. mediterraneus*, and *Pseudoalteromonas sp.*, showed a significant increase in abundance in fragments that displayed an active disease lesion. Interestingly, two of the species (*N. italica* and *T. mediteraneus*) belong to the family Rhodobacteraceae. Members of this family have diverse metabolic and physiological properties in marine systems, play important roles in the formation of marine biofilms[50–52] and are likely early colonizers at the first sign of deteriorating coral tissue[53]. The order of Rhodobacterales also has been highlighted in disease lesions of corals affected by the highly virulent stony coral tissue loss disease (SCTLD)[54]. SCTLD was not present on corals in the US Virgin Islands at the time of our study, and so the abundance of members of this order in our study may indicate that Rhodobacterales are consistently present as opportunistic colonizers in the landscape of microbial dysbiosis.

*N. italica* is known to induce bleaching in a red alga[55], and this response is temperature sensitive[55,56] leading to compromised host chemical defenses[50]. Similarly, WPD increases in prevalence in conjunction with or following thermal stress and coral bleaching events[12,44,57,58]. In this study, *N. italica* had a progressively higher relative abundance in disease-exposed and even more so in disease-infected fragments among all coral species.

*T. mediterraneus* is a Roseobacter, and is also directly implicated in Australian white syndrome of coral[59] and in diseases of lobsters[60]. *T. mediterraneus* has been observed in other Caribbean coral diseases, including black band disease[61], and is responsive to coral antibacterial activity with similar resistance as some *Vibrio* species[62]. In this study, *T. mediterraneus* had a significantly higher abundance in disease-infected fragments among all coral species.

*Pseudoalteromonas sp.*, the third bacterium detected at the highest abundance in diseased fragments also has a history of disease-associated properties and has been observed in noticeable abundance in other coral disease studies[63,64]. This bacterium is known for its antimicrobial and bacteriolytic activity in the mucus of coral and is considered to have a protective role to the coral host[65]. Interestingly, the high abundance of defensive bacteria may indicate that there was bacterial antagonism occurring. In chronic Montipora white syndrome induced by *Vibrio corallilyticus* the presence of *Pseudoalteromonas piratica* accelerated the disease because of the resistance of *Vibrio* bacteria to the antibacterial activity of *Pseudoalteromonas sp*[63] allowed undefended spread of the pathogen. This type of polymicrobial synergy with *Pseudoalteromonas sp.* altering the microbiome may allow for Rhodobacteraceae, like *N. italica* and *T. mediterraneus*, to become opportunistic pathogens[51,53] in WPD[51,53,31].

Considerations of microbial dysbiosis as a cause for marine disease have been overshadowed by a focus on identifying singular or multiple pathogens as etiological agents. Because coral disease etiological agents are so elusive, this unique dataset can be leveraged to explore the broader microbial community dysbiosis, a process that may allow bacteria to become opportunistically pathogenic[31]. Microbial dysbiosis is a microbial community shift that has a negative impact on the host and has the capacity to induce disease phenotypes[26]. Dysbiosis appears as significant microbiome shifts commonly reported in coral disease studies

**Table 2 Convergent microbiome.**

**Control**

|  | O. faveolata | O. annularis | C. natans | S. siderea | P. porites | P. astreoides | M. cavernosa |
|---|---|---|---|---|---|---|---|
| O. faveolata | - | | | | | | |
| O. annularis | 0.915 | - | | | | | |
| C. natans | 0.791 | 0.757 | - | | | | |
| S. siderea | 0.892 | 0.665 | 0.790 | - | | | |
| P. porites | 0.933 | 0.890 | 0.878 | 0.806 | - | | |
| P. astreoides | 0.955 | 0.984 | 0.940 | 0.926 | 0.910 | - | |
| M. cavernosa | 0.936 | 0.436 | 0.808 | 0.707 | 0.907 | 0.987 | - |

**Disease Exposed**

|  | O. faveolata | O. annularis | C. natans | S. siderea | P. porites | P. astreoides | M. cavernosa |
|---|---|---|---|---|---|---|---|
| O. faveolata | - | | | | | | |
| O. annularis | - | - | | | | | |
| C. natans | - | 0.924 | - | | | | |
| S. siderea | - | 0.806 | 0.707 | - | | | |
| P. porites | - | 0.916 | 0.670 | 0.688 | - | | |
| P. astreoides | - | 0.964 | 0.800 | 0.853 | 0.834 | - | |
| M. cavernosa | - | 0.381 | 0.992 | 0.875 | 0.977 | 0.993 | - |

**Disease Infected**

|  | O. faveolata | O. annularis | C. natans | S. siderea | P. porites | P. astreoides | M. cavernosa |
|---|---|---|---|---|---|---|---|
| O. faveolata | - | | | | | | |
| O. annularis | 0.928 | - | | | | | |
| C. natans | 0.710 | 0.710 | - | | | | |
| S. siderea | 0.840 | 0.673 | 0.780 | - | | | |
| P. porites | 0.797 | 0.975 | 0.802 | 0.820 | - | | |
| P. astreoides | 0.826 | 0.983 | 0.838 | 0.844 | 0.638 | - | |
| M. cavernosa | - | - | - | - | - | - | - |

The values represent the overall dissimilarity from a SIMPER analysis performed on a pairwise comparison within treatment outcome (control, disease-exposed, disease-infected) for all species. Black numbers in the control table represent the origin microbial overall dissimilarity between species. Red values indicate a microbial community that is diverging and more dissimilar than in the control conditions. Green values indicate a microbial community that is converging to be more similar than in the control conditions.

and may actually underlie many coral diseases[29,66]. Microbial dysbiosis has also been reported from thermal anomalies to parrotfish bites, indicating dysbiosis as a compounding effect from everyday scenarios on the reef[27,67].

In this study, the overall dissimilarity between treatment outcomes was compared within each species and was used to characterize dysbiosis. In orbicellids, the microbiome changed less in disease-infected fragments relative to control states compared with other species, possibly because orbicellids have a lower threshold for change before showing disease phenotypes, or etiological disease agents became pathogenic without statistically changing the microbiome assemblage. Microbial imbalance resulting from abiotic stressors could be a possible mechanism for increased susceptibility between stress events. Orbicellids are more susceptible to disease following thermal stress events[44,57,58] and our results suggest that this pattern may be due to this lower threshold for microbiome dysbiosis. Contrary to this, the disease-resistant P. astreoides showed a much higher threshold for dysbiosis in our study; their microbiome changes were significant even in fragments that were exposed to the disease but did not develop lesions. In field studies, Porites spp. are known to be tolerant to both thermal stress and subsequent disease[44], suggesting this species may tolerate a significant shift in their microbiome before holobiont break-down. This dichotomy

emphasizes how a higher dysbiosis threshold at a time of compounding environmental stressors may be an informative measurement of understanding varying stress response outcomes among coral species[27,29].

During disease exposure, stabilizing bacteria can inhibit or slow the colonization of pathogenic bacteria[68], while also preventing microbial dysbiosis. In this study, Endozoicomonas sp. were highly abundant in fragments of P. astreoides, a species that remained resistant to disease, while this bacterium was notably absent in the one fragment of this species that became diseased. In addition, the microbiome of M. cavernosa was dominated by Burkholderia ubonensis and Arthrobacter spp. within the order actinomycetales. Interestingly, both Endozoicomonas sp. and B. ubonensis increased in abundance in disease-exposed fragments, but were absent in disease-infected fragments. Endozoicomonas sp. has been largely regarded as part of the core microbiome of corals[69–71] and to be in phylosymbiosis with P. astreoides due to the coral's vertical transmission of bacteria through broadcast spawning[72]. In fact, the protective features of B. ubonensis are well known and used as a biocontrol treatment for the tropical infectious disease, meliodidosis in humans[73,74] and potentially play this role in M. cavernosa, as none of the fragments became diseased upon exposure to WPD. The dominance of these bacteria in most disease-resistant fragments suggests they may play a

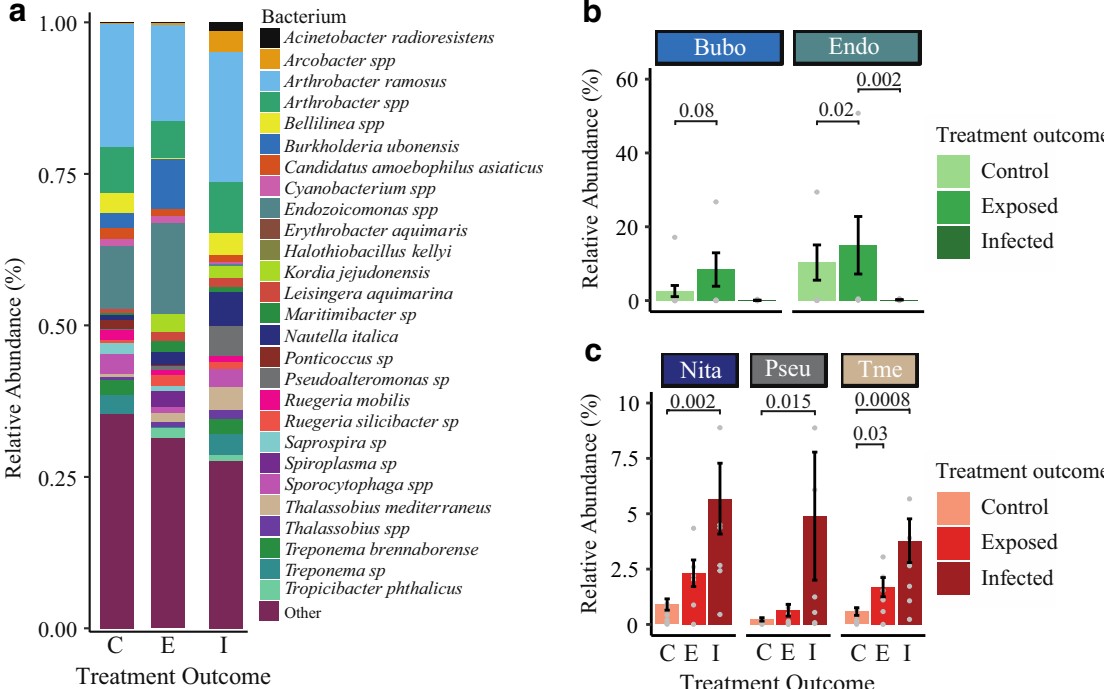

**Fig. 6 Treatment outcome-specific bacterial community. a** Bacteria of 3% relative abundance or higher in any treatment outcome. **b** Two bacteria that increased in disease-exposed fragments, (**c**) Three bacteria that significantly increased in disease-infected fragments. Abbreviations correspond to the following bacterial species: "Bubo" = *Burkholderia ubonensis*, "Endo" = *Endozoicomonas spp.*, "Nita" = *Nautella italica*, "Pseu" = *Pseudoalteromonas sp.*, "Tme" = *Thalassobius mediterraneus*. Bar plot annotations are significant *p*-values from the Tukey Post-hoc test, and the Dunn test for *Endozoicomonas spp.*, omitted *p*-value annotations are none significant comparisons. Individual data points are represented by light gray dots and error bars represent standard error.

role in the functional stabilization of the microbiome, or their commensal presence is enough to inhibit pathogenic colonization.

The increasing prevalence and severity of diseases affect coral species differently. Because the functional contributions of these species define a reef, it is integral to understand variability in disease susceptibility among coral species to predict how the disease will shape coral reefs of the future. This study characterized disease susceptibility among seven coral species representing a diversity of life-history strategies and has historically contributed to Caribbean coral reef assemblages. Less microbial change was observed in disease-susceptible coral species, suggesting low microbial dysbiosis thresholds should be further investigated as a possible coral disease etiology. Furthermore, bacteria with protective properties were more prevalent in coral species tolerant to WPD. As disease increases, disease-susceptible *Orbicella* species that are primary contributors to reef structure will become less abundant, negatively affecting the physical protection that reefs provide. This lost real estate within the reef may be colonized by more disease-resistant but less efficient reef-building species, making disease susceptibility an important predictor of the changing ecological function of Caribbean reefs.

## Methods

**Statistics and reproducibility**. This study applied a coral disease transmission methodology recently developed and reported by Williams et al. (2020). Five parental colonies from each of seven Caribbean coral species, *Orbicella faveolata, Colpophyllia natans, Siderastrea siderea, Porites astreoides, Porites porites*, and *Montastraea cavernosa*, were collected from Brewers Bay (18.34403, −64.98435), St Thomas, U.S. Virgin Islands on 13 June 2017 (Fig. 1a). Colony collection targeted either whole colonies (for *S. siderea, P. astreoides*, and *P. porites*) or fragments of colonies (for *O. faveolata, C. natans*, and *M. cavernosa*) that were between 20 and 30 cm maximum diameter. Although five parental colonies of *Orbicella annularis* were collected, one experienced mortality and was not used in experiments. Colonies were held in running seawater tables at the University of the Virgin

Islands where they were fragmented into small pieces (average size 17.74 cm² ± 1.03 SEM) using a sterilized table saw, acclimated for nine days (allowing for tissue on fragmented edges to heal completely), and then placed in experimental conditions. Diseased (*n* = 3) and healthy (*n* = 5) *O. franksi* were collected by separate divers on two dives and kept isolated from each other and from fragments of the tested coral species until the commencement of the experiment. WPD-affected colonies of *O. franksi* were identified as displaying the characteristic signs of this disease, namely large (>5 cm wide) lesions that appeared to originate from the base or edge of a colony where no signs of predation or predators were found. Diseased *O. franksi* were targeted as the source for disease in this transmission experiment because this species is known to be consistently affected by WPD throughout the year and was also used as the source species for previous experiments[19]. Diseased corals were fragmented and monitored for indications of active disease (lesion progression >0.2 cm²d⁻¹, consistent with WPD) for 24 h. Only fragments showing active lesion progression were used in disease treatments.

Upon commencement of the experiment (June 2017), coral fragments were distributed among 5 treatment and 5 control 17-L sterilized containers (17-L), each equipped with individual aerators. Containers were placed among three outdoor shaded running seawater tables that served as water baths. Containers received water changes daily and their locations were also randomized each day over the course of the 7-day experimental period. Each treatment container consisted of a randomly assigned healthy fragment of each of the seven tested species that were placed at equal distances (approximately 7–8 cm) from a central diseased *O. franksi* fragment. Control containers were identically arranged, except that healthy *O. franksi* were used as the central corals (Figs. 1b, c). During the daily water changes, the locations of the fragments within each container were randomized relative to each other, while keeping the same equal distances from the central fragment. When a disease lesion appeared on a disease-exposed coral that was previously healthy it was monitored until 30% tissue loss. If the lesion enlarged over this time period, the coral and its paired control fragment were photographed, removed, flash-frozen, and stored at −80 °C until further 16S rRNA analysis.

**Phenotype analysis**. Metrics that encompassed a susceptibility "phenotype" were calculated for each species, including: (1) total disease prevalence (% corals that became diseased) within treatment containers during the experimental period, (2) average days until lesion appearance, and (3) average lesion progression rate (cm² lost/day). Coral fragments exposed to a disease that did not show lesion appearance by the end of the transmission study were classified as "disease-exposed". While coral fragments that were exposed to the disease and developed expanding lesions were grouped as "disease-infected".

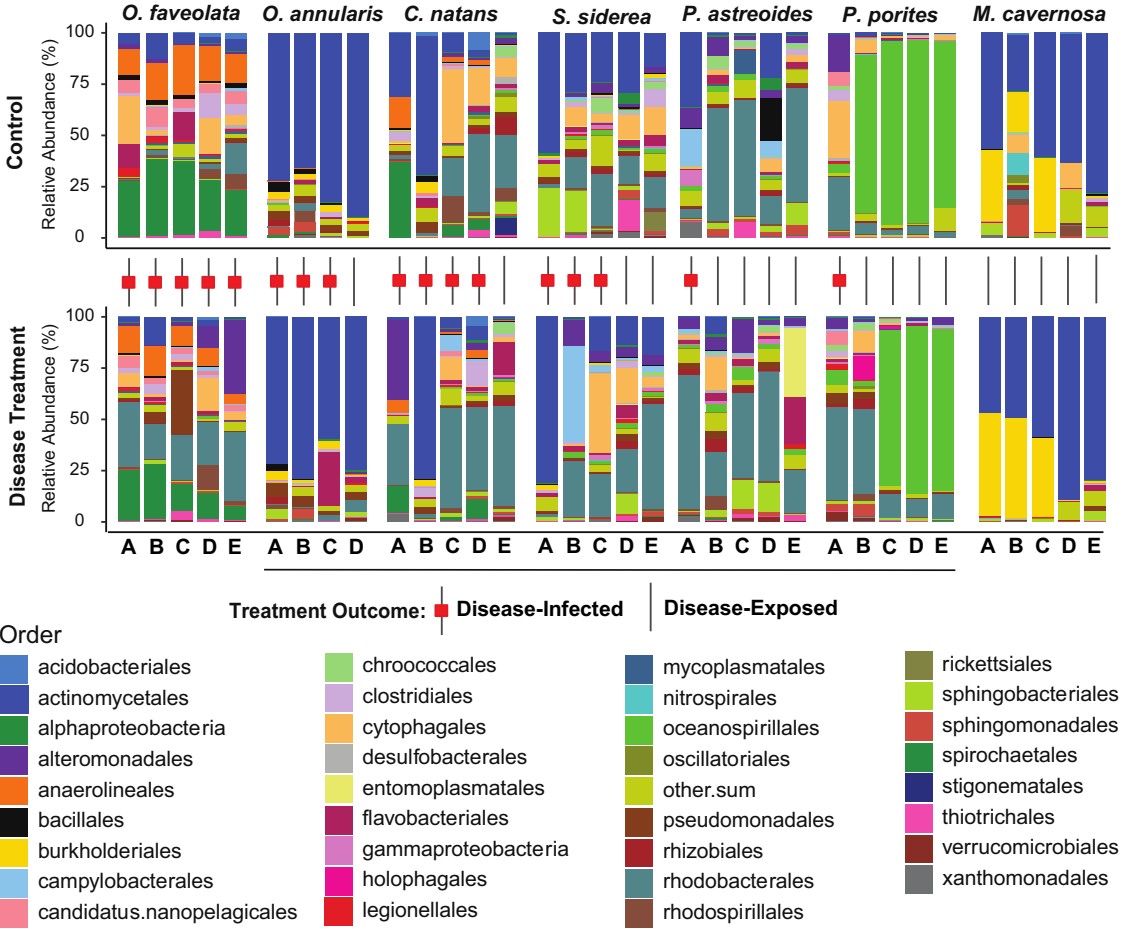

**Fig. 7 Species-specific bacterial community.** Relative abundances of bacterial orders for each coral fragment exposed to either healthy *O. franksi* (i.e., control), or a white plague-infected *O. franksi* (i.e., disease treatment). Only bacterial orders with relative abundances greater than 3% were colored individually. All other bacterial orders were grouped into the category "other.sum". Paired fragments that are genotypically identical that came from the same colony are oriented vertically to visualize how the microbiome changes from control to disease treatment exposure. The red square between vertical columns represents the genotype pair that displayed an active white plague lesion (disease-infected) by the end of the experiment.

**Table 3 Microbial diversity.**

| Kruskal-Wallis | Species | | | Species:outcome | | | Treatment | | | Outcome | | | Group | | | Spearmann to LPR | |
|---|---|---|---|---|---|---|---|---|---|---|---|---|---|---|---|---|---|
| Index | $\chi^2$ | df | $p$ | $\chi^2$ | df | $p$ | $\chi^2$ | df | $p$ | $\chi^2$ | df | $p$ | $\chi^2$ | df | $p$ | $R$ | $p$ |
| Shannon | 34.8 | 6 | <1e - 5 | 40.7 | 18 | 0.0016 | 0.048 | 1 | 0.83 | 0.24 | 2 | 0.89 | 2.97 | 2 | 0.23 | −0.14 | 0.59 |
| Pielou | 35.1 | 6 | <1e - 5 | 41.2 | 18 | 0.0013 | 0.003 | 1 | 0.95 | 0.48 | 2 | 0.78 | 4.38 | 2 | 0.11 | −0.1 | 0.69 |
| Simpson | 35.1 | 6 | <1e - 5 | 41.3 | 18 | 0.0013 | 0.007 | 1 | 0.93 | 0.26 | 2 | 0.88 | 2.87 | 2 | 0.24 | −0.06 | 0.82 |
| Beta | 21.7 | 6 | 0.0013 | 35.8 | 18 | 0.0074 | 0.471 | 1 | 0.49 | 1.37 | 2 | 0.50 | 12.4 | 2 | 0.002 | −0.21 | 0.42 |

A Kruskal-Wallis test was performed on the diversity indices. "$\chi^2$" indicates the Chi-square test, "$p$" indicates the p-value.

Disease prevalence among species was compared using Fisher's exact test in R (Supplementary Table 1). A photograph and timestamp were captured upon the appearance of lesions and then immediately before each fragment was culled around 30% tissue loss. Disease severity was measured by calculating the rate of lesion progression across the coral fragment as the amount of tissue lost between the appearance of the lesion to the time it was culled divided by that time period. Time to infection for a fragment was measured as the number of days from experiment start to the first appearance of lesions and visualized with a survival plot through a Kaplan-Meier estimate of the survivorship by using the *survfit* function in the R package *survival*[75] (Supplementary Table 2). The relative risk of each species was also calculated as:

Relative risk (RR) = Risk in exposed/Risk in non-exposed

where the *risk in exposed* individuals was calculated as the prevalence (diseased/total population) of those exposed to disease and *risk in non-exposed* individuals was calculated as the prevalence (diseased/total population) of those not exposed to the disease. To obtain an estimate of relative risk, Markov Chain Monte Carlo

simulations were used with Gibbs sampling in OpenBUGS (MRC Biostatistics Unit, Cambridge, UK). Ninety-five percent credible intervals were calculated for each estimate of relative risk. Credible intervals that did not include a value of one were considered significant, with a credible interval above one signifying a higher risk of disease because of exposure to the diseased coral fragment. A credible interval below one signified a lower risk of disease from exposure.

**Microbiome extraction and sequencing.** DNA from the coral samples was extracted at the University of Texas at Arlington (UTA) using the DNeasy Powersoil Isolation kits (MO BIO Laboratories, Carlsbad, CA). Roughly 0.25 g of tissue was removed from each of the coral fragments using a sterilized bone cutter (Supplementary Table 3). Tissue from healthy-state fragments ("control") was extracted from the center of the fragment. Tissue was extracted in a similar manner from fragments exposed to WPD that did not display lesions by the end of the experiment (disease-exposed). For fragments that developed a lesion(s) ("disease-infected"), tissue was

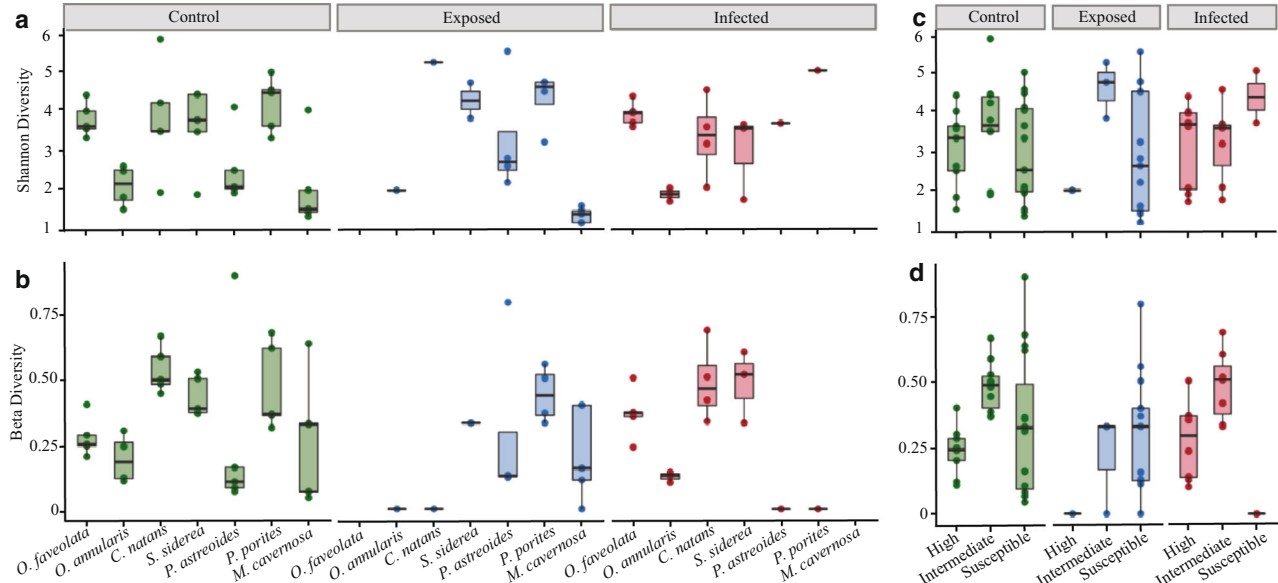

**Fig. 8 Microbiome diversity. a** Alpha diversity (Shannon diversity) and (**b**) Beta diversity (local diversity) per species and ordered by species disease susceptibility from left to right; and grouped by treatment outcome: control, exposed, and infected. **c, d** Alpha and Beta diversity grouped by species susceptibility (*O. faveolata, O. annularis* are high susceptibility, *C. natans* and *S. siderea* are intermediate susceptibilities, *P. astreoides, P. porites*, and *M. cavernosa* are in low susceptibility).

extracted approximately 2 to 3 mm horizontally from the lesion margin in the apparently healthy tissue and collected parallel to the lesion margin.

Tissue samples were sent to MR DNA Molecular Research LP (Shallowater, TX) for 16S rRNA gene amplification using 515F (GTGYCAGCMGCCGCGGTAA) and 806R (GGACTACNVGGGTWTCTAAT) primers for the V4 region and DNA libraries were prepared following MR DNA protocols. Samples were sequenced on an Illumina MiSeq 2 × 250 bp PE reads and resulting sequences were bioinformatically processed through the MR DNA pipeline (MR DNA, Shallowater, TX, USA) utilizing the QIIME analysis. Barcodes, primers, and ambiguous calls are removed from sequences, as well as short sequences <150 bp. Operational taxonomic units (OTUs) are clustered at 97% similarity and taxonomically classified using BLASTn against an RDPII and NCBI database (www.ncbi.nlm.nih.gov, http://rdp.cme.msu.edu) and organized into each taxonomic level as counts and percentage files based on industry standards of homology of sequences to the NCBI reference database (Supplementary Table 6). Counts are the number of sequences read while the percentage is the relative proportion of reads within each sample for each taxonomic classification.

**Microbial community analysis.** Differences in bacterial communities between coral species and treatment outcome levels (control, disease-exposed, disease-infected) were assessed with a two-way permutational multivariate analysis of variance (PERMA-NOVA) using the R package "vegan"[76]. Microbial community differences were visualized using non-metric multidimensional scaling (NMDS). Bacterial abundances that were most dissimilar between species and treatment outcome were identified using a similarities percentage analysis (SIMPER)[76]. These analyses were based on a Bray-Curtis dissimilarity matrix. One-hundred percent stacked bar graphs were created using raw abundance percentages of bacterial families (>3% relative abundance) to visualize shifts in the microbial composition of each individual coral and among species. Analysis of variance (ANOVA) and Tukey HSD post-hoc analyses identified differentially expressed bacteria between treatment outcomes. However, a non-parametric Kruskal-Wallis test was applied to *Endozoicomonas spp.*

The Shannon diversity index and beta diversity of the bacterial community were calculated for each sample using the R package "betapart"[77]. Diversity data were non-normal even with transformation, therefore differences among treatments were tested using non-parametric Kruskal-Wallis tests and relationships between diversity indices and lesion progression rates were investigated using non-parametric Spearman rank tests.

**Reporting summary.** Further information on research design is available in the Nature Research Reporting Summary linked to this article.

## Data availability
Source data used to create all figures including 16s sequences are stored at NCBI through BioProject accession PRJNA667272 and additionally made publicly available through the BCO-DMO project page: https://www.bco-dmo.org/project/727496.

## Code availability
Analysis for the publication was conducted in R version 3.6.2 (2019-12-12). The R scripts for the analysis are made publicly available through Github: https://doi.org/10.5281/zenodo.4635319[78].

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

## Acknowledgements

This work was supported by National Science Foundation (Biological Oceanography) award number 1712134 to L.D.M., 1712540 to M.B., and 1712240 to E.M.M, and VI EPSCoR (NSF #0814417). The authors thank the facilities and diving support staff at the University of the Virgin Islands Center for Marine and Environmental Studies (CMES), and Tyler Smith for field support. The authors acknowledge the support and consult from Kimberly Bowles at the UTA Life Sciences Genomics core facility and Scot Dowd at MR DNA. This is CMES contribution #232. Finally, we thank the two anonymous reviewers and Dr. Michael Sweet as the third reviewer for their comments, which improved this manuscript.

## Author contributions

E.M.M., L.D.M., M.B. were responsible for compiling the intellectual merit behind the research objectives, design, and acquiring funding for this original research. All authors, N.J.M., K.C., D.L., A.C.F., A.G., B.D., J.A., L.F., C.R., C.B., E.M.M., L.D.M., M.B., contributed to the experimental disease exposure. M.B. recorded phenotypic metrics, designed phenotypic analysis, and created Fig. 1a. K.C. assisted in 16S rRNA extraction and created Supplementary Fig. 1. E.M.M. designed and performed the relative risk analysis. N.J.M. designed and performed microbial analysis, performed phenotype analysis, wrote the manuscript, and created all remaining figures. N.J.M., L.D.M and M.B. had considerable editorial input during manuscript assembly with additional revisions from all authors, N.J.M., K.C., D.L., A.C.F., A.G., B.D., J.A., L.F., C.R., C.B., E.M.M., L.D.M., M.B.

## Competing interests

The authors declare no competing interests.
