## [Transparent Peer Review File · Communications Biology]

Reviewers' comments:

Reviewer #1 (Remarks to the Author):

This work is framed in a historical debt we have with coral reefs: To understand the "history" of reef bulging corals and their interaction with the whole ecosystem surroundings, we need to know more from interaction among the different organisms that should be conceived also in the functional development of the holobiont. As far as we get to know the biology of corals, we understand that the term holobiont is the clue to develop projects for answering questions about the problematic of the diseases, but also about the settlement and development. More than keep looking for ethological agents we need to focus in the homeostasis of the holobiont! And this is what I found in the background of the document. The direction of the work agrees with previous observations about dysbiosis and the symbiosis brake down that might lead the disease development.

The authors let us see that we don't have yet a clear study about coral species susceptibility and point out that it is important for generating elements in order to predict or have an idea of disease impact in coral communities. They claim that, understanding the susceptibility and the link with the microbial shifts in diseased corals, will help to understand the development of the disease, and I could not agree more.

They are supported with previous works (with references including other works published by some of the authors) in the area of immunology and distinction of phenotypes to characterize and measuring susceptibility, this makes the resulting data stronger and sustained. They could identify "susceptibility phenotypes" differences among coral species caused by WPD, they compared shifts in microbiomes and were able to relating those changes with the degree of susceptibility. That makes a strong claim for publication.

There are some publications about coral susceptibility but more related with stressors and the occurrence of disease, but not really looking the scenario of the organism failure and the problematic of the symbiont. From that point and stressing the dysbiosis component, this work has not only a novel perspective but a great potential for keeping up with more publications.

Stating clear that there are not issues with the novelty of this work and that it will be a publication of great interest for researchers in the field, I have some comments for the text.

LINE:

68, 69: The authors mention: "We further investigated individual bacteria for their potential role as disease-associated or disease preventing bacteria based on their abundances among treatments and the treatment outcomes", they point to investigate individual bacteria, in the same phrase they clarify that it is done based on their abundances. It is really problematic because the "investigation" of individuals, bacteria or others, require more than the abundances of the 16S rRNA sequences.

98: The distribution of fragments in coral containers is determining for the development of the experiment, it is important to see the positioning of species and if other factors could have any influence in the propagation of the illness, for that reasons I would suggest the authors consider to add the figure 1 of supplementary material directly on the text.

131: It would be appreciated, even if it might be obvious for current standards, if the authors specify the number of microbiome sequences they did for each coral species. Since they have three compared groups (control, exposed and infected) it is necessary to go down to the results to realise that they indeed had independent microbiomes of independent tissues for each specie in each compared group. Moreover, to mention exactly how many tissues of one species belong to one of the three groups, is more than necessary. For example, due to the susceptibility parameters, the number of tissues used to obtain the microbiome sequences, were not the same from each species: in *O. annularis* 3 were in

the group of infected and 1 in the exposed, for *C. natans* 4 and 1, and for *S. siderea* 3 and 2, respectively. To mention some. That could be confusing if it is not explained in the methodology but the differences do not risk the significance and robustness of the results. And, if it is not specifically pointed out in the methodology it could be seen as a weakness for the conclusion about compared microbiomes. We all know, therefore we understand, that number of diseased events are not possible to control!

199-219: Differences between bacterial communities and treatments, and among coral species are evident and consistent, and similarity among infected fragments (more than controls) is remarkable and it grabbed my attention (table 2). But figure 4 is, perhaps, not the best graphic to show it because dispersion is visually augmented, and colours overlap triangles and squares, making difficult to follow up what is described in the text. I would recommended a figure where one could clearly see differences and similarities. The differences among species are clear, possibly authors could show correlation using a single graphic per species.

221: When authors refer bacteria species identified (2197), the meaning of identification of bacteria must be considered. There are not sequences reported that allow to know which are those bacteria. Or they refer to reads that cover up to specie? This is something should be accurate.

223-230: The authors in figure 5, and in the text, mention 17 bacteria with significant abundance differences between treatments. These results are promising and open questions for further invaluable research, for that reason, I recommend to be clear between bacteria genus abundance and species abundances. As one can see there are 11 species mentioned and the others are at the genus level. And from those classified as species, there are not 16S rRNA sequences that could be confirmed in the reports. Accession numbers for the assignation of specie could be desirable as well as a discriminated description of the reads that reach specie level and the ones that only reach genus level. All this in order to support the specific conclusions about bacterial species.

299-299: the authors discuss that results show similarities and dissimilarities among treatments in microbial abundance, and as far as interesting and enlighten those results are, it should never be forgotten that those developments were obtained in a controlled system container.

314-316: From previous lines and after these ones (L: 333, 334), authors focus the discussion on pathogenic agents and between lines they bring out the search for an etiological agent for the WPD, or for an individual responsible of the damage caused by its pathogenic teats. I think that's not necessary, more if they really started the paper talking about dysbiosis, even more if they mention all the immunological response from the coral that also might be involved in the disease development, even more when they bring from references that we might not know if the illness is a consequence or a cause for the microbial shift (L: 351-353).

354: From this line and further on, the authors discuss how low threshold for microbial dysbiosis could have a role in coral susceptibility. The immunological system and response of corals and the immunological "agreement" with symbiont might have an important responsibility, as it is mention in the lines and I think if they, additionally, make a stronger argument using references as Palmer CV, Traylor-Knowles N (2012) *Proc Biol Sci* 279:4106–4114, Mydlarz LD, (2009) *Dis Aquat Organ* 87:67–78, Ocampo-Ibáñez, I (2015). *Immunogenetics*. 67, it could make this discussion even superior. But this is not necessary it is already strong enough. It might be a topic for another work from this team.

Reviewer #2 (Remarks to the Author):

This study aims to investigate the relationship between the coral microbiome and disease susceptibility, specifically White Plague Disease, in an effort to better understand potential

mechanisms that could be driving disease resistance. To do this, the authors infected corals in a laboratory setting and examined differences in phenotypic and microbial responses to WPD. Acknowledging that previous studies have shown disease susceptibility and resistance to vary between species, the authors selected 7 species with varied life-history strategies and ecosystem roles for their experiment. Given the relevance of this study as well as its solid design and results, I recommend it for publication with some minor edits.

The experimental design used as well as the data analysis methods employed are sound and appropriate for answering the questions presented in the introduction of this manuscript. The data presented clearly and concisely, and the figures are clear as well. There are some things the authors should change or add in the methods, results, and discussion sections:

Methods:

I believe it worth mentioning that the description of the experimental design can get a bit confused, and it would be helpful for readers to have a diagram with the layout of the design (ie. fragment spread across tanks/#s of fragments and fragment types per tank, # treatment tanks vs control, etc).

I also recommend that the authors add details regarding the processing pipeline used by MR, for example whether the sequences were processed using mothur or QIIME or DADA2, whether they are using OTUs or ASVs, the database that was used to assign taxonomic information to the sequences, etc.

Discussion:

I was a little confused regarding the implications of your section on dysbiosis. The beginning of this section (line 344) starts with the discussion of microbial dysbiosis as a cause of disease. However, is there not something that ultimately has to cause that dysbiosis, then spread to other corals? In the case of this experiment, by placing an infected coral in a tank with other fragments, the other fragments developed the same disease. This suggests that there is some sort of disease vector at play. In this case, it is possible that the dysbiosis is directly causing the disease, but would there not have to be some other agent that causes the dysbiosis? In which case, dysbiosis could be considered an intermediary step between disease vector and disease. The alternative is the idea that a coral might be spreading its dysbiosis to other corals. I think regardless it would be worth discussing spreadability (i.e., is the diseased coral spreading microbiome-disrupting bacteria or releasing microbiome-disrupting compounds into the water column?) if we're going to consider microbiome dysbiosis as the causative agent of disease. Apologies if I am misinterpreting or misunderstanding, it's an interesting idea!

Data deposition/reproducibility:

If they haven't already, I ask that the authors submit their sequence data to a database to be made publicly available. I also recommend that the authors upload the scripts used to process the data and generate figures to a Github repository to be made publicly available, if this has not been done yet.

General comments/questions:

57: Remove apostrophe

83: Is a 9-day period of acclimation enough to ensure that the corals are not experiencing negative effects from the fragmentation during the course of the experiment?

142: Redundant sentence, in the previous sentence the primers have already been listed.

209: remove "that"

295: Is it possible that we're seeing similar patterns of colonization by opportunistic bacteria as well?

Reviewer #3 (Remarks to the Author):

The study by MacKnight et al explores disease resistance across a number of coral species in the Caribbean.

Something which is not so clear to me from the start is how the authors separate out WPD from SCTL for example the line 43 would apply to both? In my view I wouldn't be surprised if its actually the same disease or group of diseases but to avoid confusion clear differences should be outlined and evidence provided where necessary.

Line 52 where you hint on questioning the view of singular pathogenic etiology you should probably cite these two studies

<https://www.frontiersin.org/articles/10.3389/fmars.2017.00009/full>

<https://link.springer.com/article/10.1186/s40168-019-0759-6>

both of which strongly suggest this to be the case for all diseases

Line 53, seems a little odd, the word dysbiosis might not have been mentioned in all studies but it is only a term encapsulating whats happening, so the actual shift/change has actually been shown in all instances of diseases – suggest removing or changing this line to reflect as such

L58 I think this concept is widely acknowledged in coral disease literature just not maybe under the term dysbiosis – again see the review

<https://www.frontiersin.org/articles/10.3389/fmars.2017.00009/full>

The concept of an early warning system by following the microbial communities has also been indicated a number of times – see again <https://link.springer.com/article/10.1186/s40168-019-0759-6>

For a coral disease study highlighting exactly this and also references withing <https://osf.io/gv6s7>

Also see Rorder et al. 2015, Zanevald et al. 2017, Glasl et al. 2019, Fan et al. 2013 referenced in the above preprint

See also <https://osf.io/gv6s7> for future management tools which discusses some aspects related to this study around the identification of individual colonies with disease resistance for example Methods – would be useful to know the sizes of the colonies and resulting fragments to link this to the progression rates – I'm presuming they were large to account for some of the faster disease progression rates indicated in the results section. How do these rates relate to those monitored in the field. This is an age old issue with coral disease work, i.e. ensuring the disease you are working on is infact the disease you identify in the field and not what aquarists refer to as rapid tissue necrosis or some such other loss of tissue resulting issue

In the intro L69 you indicate that you investigated individual bacteria for their potential role as disease-associated or disease preventing bacteria – this implied to me you did challenge experiments rather than just looking for trends in the microbiome data – I would personally alter this and/or not even really draw attention to this analysis as this is routine in many disease studies. Further such analysis is speculative and has no real value with regard to identification of likely pathogens – especially with such few replicates in the case of this study. A mistake many disease studies have tried to do and ultimately failed

L292 – I'm not so convinced saying that the clustering of the infected colonies suggests they have the same disease – more likely that similar opportunistic bacteria take over as you have them all in the same environment/tank and so a homogenization of the pathobiome is to be expected. The only way in my view to show it's the same disease, is to show the same pathogen/pathogens consistently present in the various colonies, which you can not do and which no one has been able to consistently

do – hence the argument that the ‘disease’ is either multiple disease effecting different species in different ways (likely) or that the ‘disease’ is caused by any number of opportunistic bacteria or multiples of – (also highly likely) – regardless – you haven’t proven any of this in your study – so I would tone these statements down in my opinion.

L300 – what size fragments did you receive from sequencing? I doubt they were suitable to identify bacteria to species level – this is another common mistake seen in many coral papers – many species of bacteria, especially pathogenic species are notoriously difficult to identify with 16S alone as well – hence the use of a number of house keeping genes are often utilised to confirm or deny ID – see <https://onlinelibrary.wiley.com/doi/abs/10.1111/mec.13326> - this might be worthy of discussion. .

L310 and the arguments above seem to point to this disease not being WP but more closely related to SCTL D – if you put credence on the ID of pathogens which you seem to do – however this is my point, it’s a slippery slope to go down and best avoided as you have no evidence to point one way or the other – WE DO NOT KNOW THE CAUSAL AGENTS OF ANY CORAL DISEASE (without any responsible doubt that is) – why do most people still try and find a single pathogen??? L333 – why is it so hard to believe that the concept of the one pathogen = one disease does not work for corals see <https://www.frontiersin.org/articles/10.3389/fmars.2017.00009/full>. Every single paper which comes out provides more and more evidence to support this idea over the concept of known single pathogens, yet people still write or try and find a single pathogen. It baffles me. L341, no, no, no – no evidence – please remove as this sort of speculation continues to confuse people!!!

L344, yes much better but again this has been suggested before – see previous studies not cited in this paper.

When discussing dissimilarity you could refer to this latest study as well

<https://link.springer.com/article/10.1186/s40168-019-0759-6>

Nice section on Endozoicomonas

Signed - Michael Sweet

Our responses to reviewer's comments can be found in **bold** below each comment.

Referee expertise:

Referee #1: coral host-microbe interactions

Referee #2: coral microbiology

Referee #3: coral reef epidemiology, microbial pathogens

Reviewers' comments:

Reviewer #1 (Remarks to the Author):

This work is framed in a historical debt we have with coral reefs: To understand the “history” of reef bulging corals and their interaction with the whole ecosystem surroundings, we need to know more from interaction among the different organisms that should be conceived also in the functional development of the holobiont. As far as we get to know the biology of corals, we understand that the term holobiont is the clue to develop projects for answering questions about the problematic of the diseases, but also about the settlement and development. More than keep looking for ethological agents we need to focus in the homeostasis of the holobiont! And this is what I found in the background of the document. The direction of the work agrees with previous observations about dysbiosis and the symbiosis brake down that might lead the disease development.

The authors let us see that we don't have yet a clear study about coral species susceptibility and point out that it is important for generating elements in order to predict or have an idea of disease impact in coral communities. They claim that, understanding the susceptibility and the link with the microbial shifts in diseased corals, will help to understand the development of the disease, and I could not agree more.

They are supported with previous works (with references including other works published by some of the authors) in the area of immunology and distinction of phenotypes to characterize and measuring susceptibility, this makes the resulting data stronger and sustained. They could identify “susceptibility phenotypes” differences among coral species caused by WPD, they compared shifts in microbiomes and were able to relating those changes with the degree of susceptibility. That makes a strong claim for publication.

There are some publications about coral susceptibility but more related with stressors and the occurrence of disease, but not really looking the scenario of the organism failure and the problematic of the symbiont. From that point and stressing the dysbiosis component, this work has not only a novel perspective but a great potential for keeping up with more publications.

Stating clear that there are not issues with the novelty of this work and that it will be a publication of great interest for researchers in the field, I have some comments for the text.
The authors thank this reviewer for their thoughtful analysis and review of our manuscript.

LINE:

68, 69: The authors mention: “We further investigated individual bacteria for their potential role as disease-associated or disease preventing bacteria based on their abundances among treatments and the treatment outcomes”, they point to investigate individual bacteria, in the same phrase they clarify that it is done based on their abundances. It is really problematic because the “investigation” of individuals, bacteria or others, require more than the abundances of the 16S rRNA sequences.

Yes, we agree it gave the wrong impression of what work was done.

To address this comment the following changes were made in the introduction on lines 67-63: We removed “We investigated” and changed to “We then referred to the literature for any known functional roles and relevance in coral studies of” which more accurately reflects the work fulfilled.

98: The distribution of fragments in coral containers is determining for the development of the experiment, it is important to see the positioning of species and if other factors could have any influence in the propagation of the illness, for that reason I would suggest the authors consider to add the figure 1 of supplementary material directly on the text.

We agree that the distribution of fragments in the containers could influence the development of the experiment. However, we randomized the position of the fragments each day during the water changes. We have added the following text to the methods section on lines 102-104 to clarify this: “During the daily water changes, the locations of the fragments within each container were randomized relative to each other, while keeping the same equal distances from the central fragment.” We do support the idea of adding the experimental figures to the main text and as such have redesigned figure 1 so it shows the representative experimental set-up for BOTH the control and experimental tanks and pictures of the coral in the wild.

131: It would be appreciated, even if it might be obvious for current standards, if the authors specify the number of microbiome sequences they did for each coral species. Since they have three compared groups (control, exposed and infected) it is necessary to go down to the results to realise that they indeed had independent microbiomes of independent tissues for each species in each compared group.

Moreover, to mention exactly how many tissues of one species belong to one of the three groups, is more than necessary. For example, due to the susceptibility parameters, the number of tissues used to obtain the microbiome sequences, were not the same from each species: in *O. annularis* 3 were in the group of infected and 1 in the exposed, for *C. natans* 4 and 1, and for *S. siderea* 3 and 2, respectively. To mention some. That could be confusing if it is not explained in the methodology but the differences do not risk the significance and robustness of the results. And, if

it is not specifically pointed out in the methodology it could be seen as a weakness for the conclusion about compared microbiomes. We all know, therefore we understand, that number of diseased events are not possible to control!

Yes, we have added a more direct reference to how many tissues and microbiomes were sequenced and this will bring clarity to the treatment outcome and microbial sequencing. Therefore we have added a Table to the Supplementary information that includes the number of fragments that were controls, disease-exposed, or disease-infected treatment with the outcomes. Supplementary Table 3, referenced in the methods on line 139 and provided below.

	O. faveolata	O. annularis	C. natans	S. siderea	P. porites	p. astreoides	M. cavernosa	Total
C	5	4	5	5	5	5	5	34
DE	0	1	1	2	4	4	5	17
DI	5	3	4	3	1	1	0	17
Total	10	8	10	10	10	10	10	68

199-219: Differences between bacterial communities and treatments, and among coral species are evident and consistent, and similarity among infected fragments (more than controls) is remarkable and it grabbed my attention (table 2). But figure 4 is, perhaps, not the best graphic to show it because dispersion is visually augmented, and colours overlap triangles and squares, making difficult to follow up what is described in the text. I would recommended a figure where one could clearly see differences and similarities. The differences among species are clear, possibly authors could show correlation using a single graphic per species.

Yes, we agree with the reviewer’s comment. To address the comment, we referenced table 2 in the results on lines 218, 226, and 306 which better represents differences in treatment outcomes. We also referenced the NMDS in figure 5 (previously figure 4) to statements that illustrate the species differences in the results on line 212 and 229 which better fit the figure.

221: When authors refer bacteria species identified (2197), the meaning of identification of bacteria must be considered. There are not sequences reported that allow to know which are those bacteria. Or they refer to reads that cover up to specie? This is something should be accurate.

To address this comment we have changed the text to:

Line 232-234: “Of the 7,225 unique OTUs, 1,243 OTUs had greater than 97% sequence similarity to reference sequences allowing for species level classification of 1,243 unique bacterial species identified in this study...”

Additionally, Supplementary Table 6 was added and referenced in Line 155. This Supplementary Figure 5 provides the percent homology cutoff between our sequences and reference sequences required to assign the various levels of taxonomic classification.

223-230: The authors in figure 5, and in the text, mention 17 bacteria with significant abundance differences between treatments. These results are promising and open questions for further invaluable research, for that reason, I recommend to be clear between bacteria genus abundance and species abundances. As one can see there are 11 species mentioned and the others are at the genus level. And from those classified as species, there are not 16S rRNA sequences that could be confirmed in the reports. Accession numbers for the assignation of specie could be desirable as well as a discriminated description of the reads that reach specie level and the ones that only reach genus level. All this in order to support the specific conclusions about bacterial species.

We agree that the species and genus level classification abundance data must be respected and have adjusted the taxonomic classification language to the genus level.

Figure 6A (previously Figure 5A) was originally created from our dataset and included some eukaryotic phyla. We removed all the extraneous phyla and now Figure 6A properly represents exclusively prokaryotes. The objective of this figure is to bring attention to and consider bacteria of highest relative abundance from the disease transmission study. In doing so, we considered bacteria greater than 1% relative abundance in any treatment outcome rather than bacteria of 3% or higher relative abundance in any treatment outcome as originally performed. This revised analysis provided 29 bacteria at the genus level that consisted of approximately 70% of the total microbiome abundance as referenced in Line 239.

299-299: the authors discuss that results show similarities and dissimilarities among treatments in microbial abundance, and as far as interesting and enlighten those results are, it should never be forgotten that those developments were obtained in a controlled system container.

We completely understand this point and feel our methods and new figure 1 will enhance this point, referenced in the Methods on line 75,101. It should also be mentioned that without doing this experiment in mesocosms, we would never have the resolution to separate corals that were exposed but never got diseased and we are able to catch lesion growth in the first few hours of development. This is nearly impossible to achieve in the wild.

314-316: From previous lines and after these ones (L: 333, 334), authors focus the discussion on pathogenic agents and between lines they bring out the search for an etiological agent for the WPD, or for an individual responsible of the damage caused by its pathogenic teats. I think that's not necessary, more if they really started the paper talking about dysbiosis, even more if they mention all the immunological response from the coral that also might be involved in the disease development, even more when they bring from references that we might not know if the illness is a consequence or a cause for the microbial shift (L: 351-353).

We agree that dedicating a lot of space to the search for an etiological agent is unproductive, especially given the state of the coral disease field and the focus on one pathogen. We truly appreciate this reviewer's comments as we didn't quite realize how much verbiage we had dedicated to this. We have reframed that discussion to highlight the

3 differentially abundant bacteria we found in our dataset, which now leads into our discussion of dysbiosis. We are also excited to see the reviewer’s enthusiastic response to our discussion of microbial dysbiosis and we look forward to furthering this discourse.

354: From this line and further on, the authors discuss how low threshold for microbial dysbiosis could have a role in coral susceptibility. The immunological system and response of corals and the immunological “agreement” with symbiont might have an important responsibility, as it is mentioned in the lines and I think if they, additionally, make a stronger argument using references as Palmer CV, Traylor-Knowles N (2012) Proc Biol Sci 279:4106–4114, Mydlarz LD, (2009) Dis Aquat Organ 87:67–78, Ocampo-Ibáñez, I (2015). Immunogenetics. 67, it could make this discussion even superior. But this is not necessary it is already strong enough. It might be a topic for another work from this team.

The authors thank the reviewer for this, and the citations were included but after further conversation, the sentence that was cited was removed from the discussion.

Reviewer #2 (Remarks to the Author):

This study aims to investigate the relationship between the coral microbiome and disease susceptibility, specifically White Plague Disease, in an effort to better understand potential mechanisms that could be driving disease resistance. To do this, the authors infected corals in a laboratory setting and examined differences in phenotypic and microbial responses to WPD. Acknowledging that previous studies have shown disease susceptibility and resistance to vary between species, the authors selected 7 species with varied life-history strategies and ecosystem roles for their experiment. Given the relevance of this study as well as its solid design and results, I recommend it for publication with some minor edits.

The authors thank this reviewer for the positive comments.

The experimental design used as well as the data analysis methods employed are sound and appropriate for answering the questions presented in the introduction of this manuscript. The data presented clearly and concisely, and the figures are clear as well. There are some things the authors should change or add in the methods, results, and discussion sections:

Methods:

I believe it worth mentioning that the description of the experimental design can get a bit confused, and it would be helpful for readers to have a diagram with the layout of the design (ie. fragment spread across tanks/#s of fragments and fragment types per tank, # treatment tanks vs control, etc).

We value this observation and agree it will bring clarity to the reader in regard to experimental design. In our comments and responses to all the reviewer’s questions, we have added a table consisting of the number of fragments from each species that ended up in each treatment outcome at the conclusion of the study (Supplementary Table 3 and referenced in Line 139). Furthermore, in the caption of Supplementary Figure 1 we added

on line 641, “There were five aquaria assigned as control treatments and five aquaria assigned as disease treatments. Every tank included one fragment from each of the seven coral species.” We have also added the experimental design figures to the main text.

I also recommend that the authors add details regarding the processing pipeline used by MR, for example whether the sequences were processed using mothur or QIIME or DADA2, whether they are using OTUs or ASVs, the database that was used to assign taxonomic information to the sequences, etc.

To address this comment we have edited the text in the methods to be:

Line 149-156: “...utilizing the QIIME analysis. Barcodes, primers, and ambiguous calls are removed from sequences as well as short sequences <150 bp. Operational taxonomic units (OTUs) are clustered at 97% similarity and taxonomically classified using BLASTn against an RDP-II and NCBI database (www.ncbi.nlm.nih.gov, <http://rdp.cme.msu.edu>) and organized into each taxonomic level as counts and percentage files. Counts are the number of sequences read while percentage is the relative proportion of reads within each sample for each taxonomic classification.”

Discussion:

I was a little confused regarding the implications of your section on dysbiosis. The beginning of this section (line 344) starts with the discussion of microbial dysbiosis as a cause of disease. However, is there not something that ultimately has to cause that dysbiosis, then spread to other corals? In the case of this experiment, by placing an infected coral in a tank with other fragments, the other fragments developed the same disease. This suggests that there is some sort of disease vector at play. In this case, it is possible that the dysbiosis is directly causing the disease, but would there not have to be some other agent that causes the dysbiosis? In which case, dysbiosis could be considered an intermediary step between disease vector and disease. The alternative is the idea that a coral might be spreading its dysbiosis to other corals. I think regardless it would be worth discussing spreadability (i.e., is the diseased coral spreading microbiome-disrupting bacteria or releasing microbiome-disrupting compounds into the water column?) if we're going to consider microbiome dysbiosis as the causative agent of disease. Apologies if I am misinterpreting or misunderstanding, it's an interesting idea!

We agree and have rewritten a lot of this part of the discussion to go a bit deeper into the dysbiosis hypothesis as well as the major taxon that showed differential abundance among treatments.

Data deposition/reproducibility:

If they haven't already, I ask that the authors submit their sequence data to a database to be made publicly available. I also recommend that the authors upload the scripts used to process the data and generate figures to a Github repository to be made publicly available, if this has not been done yet.

Data Availability is now stated on Line 414. The microbiome sequences have been submitted to NCBI and will be made available additionally at the BCO-DMO Project Page url: <https://www.bco-dmo.org/project/727496>

The R scripts used for this manuscript's analysis has been made publicly available on the Github Code Repository url: <https://github.com/nmacknight/16sCommunityAnalysis>

General comments/questions:

57: Remove apostrophe

Removed.

83: Is a 9-day period of acclimation enough to ensure that the corals are not experiencing negative effects from the fragmentation during the course of the experiment?

In previous experiments, it was determined that 9 days was sufficient for the tissue on the edges of the cut parts of the coral to heal within this timeframe. We have added the following to text in the methods:

(Lines 79-83) – “Colonies were held in running seawater tables at the University of the Virgin Islands where they were fragmented into small pieces (average size $17.74 \text{ cm}^2 \pm 1.03$ SEM) using a sterilized table saw, acclimated for nine days (allowing for tissue on fragmented edges to heal completely), and then placed in experimental conditions.” In addition, control fragments showed no ill effects from fragmentation, including no development of lesions. We feel that any potential negative effects from fragmentation that were not observable had little to no influence on the experimental outcomes.

142: Redundant sentence, in the previous sentence the primers have already been listed.
Fixed and reformatted.

209: remove “that”

Removed.

295: Is it possible that we're seeing similar patterns of colonization by opportunistic bacteria as well?

Yes, certainly. We agree and before annotating any bacteria with a functional role such as opportunistic we begin by observing that the microbiome is more similar in a disease-infected state than any other state. Thereafter we can refer to the literature on these communities for their ecological properties.

Reviewer #3 (Remarks to the Author):

The study by MacKnight et al explores disease resistance across a number of coral species in the Caribbean.

Something which is not so clear to me from the start is how the authors separate out WPD from

SCTLD for example the line 43 would apply to both? In my view I wouldn't be surprised if its actually the same disease or group of diseases but to avoid confusion clear differences should be outlined and evidence provided where necessary.

Our experiment took place in June 2017 (detailed in methods) and SCTLD was not identified on US Virgin Islands reefs at that time. Its first appearance in the USVI was well documented and occurred in January of 2019. We have added clarifying text that we believe addresses this potential confusion.

This included adding additional details to line 43 (now lines 40-44) explaining the unique characteristics of WPD: “WPD has been described affecting Caribbean corals since the 1970s and is characterized by lesions originating at the base of the colony and expanding rapidly, resulting in significant partial and total mortality to affected colonies (17). WPD is a suspected bacterial infection (17), though there has been considerable debate as to whether WPD represents one or more etiologies (18).”

We have also added to the Methods (lines 85-88) the following text: “WPD-affected colonies of *O. franksi* were identified as displaying the characteristic signs of this disease, namely large (> 5 cm wide) lesions that appeared to originate from the base or edge of a colony where no signs of predation or predators were found.”

We have modified text in the methods (lines 90-92) to the following: “Diseased corals were fragmented and monitored for indications of active disease (lesion enlargement > 0.2 cm²d⁻¹, consistent with WPD) for 24 hours.”

Line 52 where you hint on questioning the view of singular pathogenic etiology you should probably cite these two studies both of which strongly suggest this to be the case for all diseases.
<https://www.frontiersin.org/articles/10.3389/fmars.2017.00009/full>
<https://link.springer.com/article/10.1186/s40168-019-0759-6>

Citations added to line 53.

Line 53, seems a little odd, the word dysbiosis might not have been mentioned in all studies but it is only a term encapsulating whats happening, so the actual shift/change has actually been shown in all instances of diseases – suggest removing or changing this line to reflect as such.
AND

L58 I think this concept is widely acknowledged in coral disease literature just not maybe under the term dysbiosis – again see the review <https://www.frontiersin.org/articles/10.3389/fmars.2017.00009/full>

Yes, we agree. We have rewritten this paragraph to emphasize less the number of references and more about the observation of microbial shifts and dysbiosis. To address the comment the following change was made in lines 52-54. “In coral diseases, microbiome shifts or dysbiosis also may be more appropriate than the one-pathogen-one disease concept (24,25,27–33).”

The concept of an early warning system by following the microbial communities has also been indicated a number of times – see again <https://link.springer.com/article/10.1186/s40168-019-0759-6>

For a coral disease study highlighting exactly this and also references withing <https://osf.io/gv6s7>

Also see Rorder et al. 2015, Zanevald et al. 2017, Glasl et al. 2019, Fan et al. 2013 referenced in the above preprint

See also <https://osf.io/gv6s7> for future management tools which discusses some aspects related to this study around the identification of individual colonies with disease resistance for example

Citations added to lines 53,57.

Methods – would be useful to know the sizes of the colonies and resulting fragments to link this to the progression rates – I’m presuming they were large to account for some of the faster disease progression rates indicated in the results section. How do these rates relate to those monitored in the field. This is an age old issue with coral disease work, i.e. ensuring the disease you are working on is infact the disease you identify in the field and not what aquarists refer to as rapid tissue necrosis or some such other loss of tissue resulting issue

To clarify the sizes of the colonies and fragments we have added the following text to the Methods section:

Lines 75-78: “Colony collection targeted either whole colonies (for *S. siderea*, *P. astreoides*, and *P. porites*) or fragments of colonies (for *O. faveolata*, *C. natans*, and *M. cavernosa*) that were between 20 and 30 cm maximum diameter.”

Lines 79-83 were edited to: “Colonies were held in running seawater tables at the University of the Virgin Islands where they were fragmented into small pieces (average size $17.74 \text{ cm}^2 \pm 1.03 \text{ SEM}$) using a sterilized table saw, acclimated for nine days, and then placed in experimental conditions.”

Lesion progression rates for WPD from the field have been published for linear rates ($0.23 \text{ cm d}^{-1} \pm 0.12 \text{ SE}$, Brandt et al. 2013) but not for areal rates as we have presented here. Areal rates more accurately reflect the loss of tissue in the experiment and have been used more frequently in the recent coral disease literature, including for experiments in the US Virgin Islands ($0.8 \text{ cm}^2 \text{ d}^{-1} \pm 0.2 \text{ SE}$). Our areal rates were highly variable (range: $0.5 - 23.7 \text{ cm}^2 \text{ d}^{-1}$), reflecting the differences among species, but with a range that overlapped this previously reported areal rate.

If lesions were the result of aquarium-related issues, such as rapid tissue necrosis, we would expect to see such issues appear in the control containers and not just the disease treatment containers. However, no lesions were apparent on corals in the control containers over the course of the experiment. Furthermore, several coral fragments in the disease containers did not get diseased or apparent lesions during the time of our study, therefore if it was an aquarium-related issue we expect to see disease on all fragments.

In the intro L69 you indicate that you investigated individual bacteria for their potential role as disease-associated or disease preventing bacteria – this implied to me you did challenge experiments rather than just looking for trends in the microbiome data – I would personally alter this and/or not even really draw attention to this analysis as this is routine in many disease studies. Further such analysis is speculative and has no real value with regard to identification of likely pathogens – especially with such few replicates in the case of this study. A mistake many disease studies have tried to do and ultimately failed

Yes, we agree this verbiage was misleading and in response to reviewer 1 as well, we modified the statement to “We then referred to the literature for any known functional roles and relevance in coral studies of” on lines 63-66 which more accurately reflects the work fulfilled.

L292 – I’m not so convinced saying that the clustering of the infected colonies suggests they have the same disease – more likely that similar opportunistic bacteria take over as you have them all in the same environment/tank and so a homogenization of the pathobiome is to be expected.

Yes, we understand this concern. What perhaps wasn’t clear in the methods was that each infected colony was removed from the tanks at a different time and depending on the lesion growth rate, this time wasn’t the same among all coral fragments or species. We think with this experimental design, getting some similarity among the microbiomes of all species in the infected tissue is quite interesting. We have edited/added the following text in the Discussion section to address this comment and provide clarity:

(Lines 306-314) “This response suggests diseased fragments had similar microbial constituents while in the diseased state, and this could reflect a community of pathogenic or opportunistic bacteria. This microbial convergence is likely the result of disease exposure rather than tank effects because diseased fragments started showing signs of tissue loss at different time points in the study and were removed from tanks based on differing fragment lesion development. In addition, the microbiomes of fragments that were exposed to disease, but stayed apparently healthy became more dissimilar than in the controlled state. To further explore the convergent microbiome in disease-infected fragments, bacteria increasing in relative abundance in this disease-infected state were identified.”

The only way in my view to show it’s the same disease, is to show the same pathogen/pathogens consistently present in the various colonies, which you can not do and which no one has been able to consistently do – hence the argument that the ‘disease’ is either multiple disease effecting different species in different ways (likely) or that the ‘disease’ is caused by any number of opportunistic bacteria or multiples of – (also highly likely) – regardless – you haven’t proven any of this in your study – so I would tone these statements down in my opinion.

Ok, we understand this point. We have edited the Discussion section in response to this comment. This included editing and adding text mentioned in the previous comment (Lines 306-314), as well as adding the following statements: (Lines 320-324) “The order of

Rhodobacterales also has been highlighted in disease lesions of corals affected by the highly virulent stony coral tissue loss disease (SCTLD) (57). SCTLD was not present on corals in the US Virgin Islands at the time of our study, and so the abundance of members of this order in our study may indicate that Rhodobacterales are consistently present as opportunistic colonizers in the landscape of microbial dysbiosis.

L300 – what size fragments did you receive from sequencing? I doubt they were suitable to identify bacteria to species level – this is another common mistake seen in many coral papers – many species of bacteria, especially pathogenic species are notoriously difficult to identify with 16S alone as well – hence the use of a number of house keeping genes are often utilised to confirm or deny ID – see <https://onlinelibrary.wiley.com/doi/abs/10.1111/mec.13326> - this might be worthy of discussion.

To address this comment, we added Supplementary Table 6 of the percent homology used to identify sequences at each taxonomic level and is referenced in Line 155. Fragments 290bp long of the 16s V4 region using 515 forward and 806 reverse primers were sequenced which produced 1,243 operational taxonomic units (OTUs) with greater than 97% sequence homology to the NCBI reference sequence which qualified for species level classification.

L310 and the arguments above seem to point to this disease not being WP but more closely related to SCTLD – if you put credence on the ID of pathogens which you seem to do – however this is my point, it’s a slippery slope to go down and best avoided as you have no evidence to point one way or the other – WE DO NOT KNOW THE CAUSAL AGENTS OF ANY CORAL DISEASE (without any responsible doubt that is) – why do most people still try and find a single pathogen???

AND

L333 – why is it so hard to believe that the concept of the one pathogen = one disease does not work for corals see <https://www.frontiersin.org/articles/10.3389/fmars.2017.00009/full>. Every single paper which comes out provides more and more evidence to support this idea over the concept of known single pathogens, yet people still write or try and find a single pathogen. It baffles me.

We agree that dedicating a lot of space to the search for an etiological agent is unproductive, especially given the state of the coral disease field and the focus on one pathogen. We truly appreciate this reviewers comments and those of reviewer one above, as we didn’t quite realize how much verbiage we had dedicated to this. We have reframed that discussion to highlight the 3 highest differentially abundant bacteria we found in our dataset and have that lead into our discussion of dysbiosis. We have also added a comment regarding SCTLD. Since this disease was obviously not SCTLD, we see value in relating the involvement of certain bacterial families to both diseases and their potential opportunistic nature.

In order to address this comment we have edited the text in the Discussion section to: (Lines 320-324) “The order of Rhodobacterales also has been highlighted in disease lesions of corals affected by the highly virulent stony coral tissue loss disease (SCTLD) (57).

SCTLD was not present on corals in the US Virgin Islands at the time of our study, and so the abundance of members of this order in our study may indicate that Rhodobacterales are consistently present as opportunistic colonizers in the landscape of microbial dysbiosis.”

L341, no, no, no – no evidence – please remove as this sort of speculation continues to confuse people!!!

We have removed this part of the text.

L344, yes much better but again this has been suggested before – see previous studies not cited in this paper.

To address the comment, we have included the provided and valued citations.

When discussing dissimilarity you could refer to this latest study as well <https://link.springer.com/article/10.1186/s40168-019-0759-6>

To address the comment, we have included the provided and valued citations.

Nice section on Endozoicomonas.

Signed - Michael Sweet

REVIEWERS' COMMENTS:

Reviewer #1 (Remarks to the Author):

Congratulation to the authors, I see the improvement of the document. I have no further comments, just add that the final phrase: "This lost real estate within the reef may be colonized by more disease resistant but less efficient reef-building species, making disease susceptibility an important predictor of the changing ecological function of Caribbean reefs.", is spicy and brave. I liked it!

Reviewer #2 (Remarks to the Author):

The authors have addressed all of my comments and suggestions and have provided an improved version of the manuscript. I recommend this manuscript for publication.